# OpenReview forum: "SoSBench: Benchmarking Safety Alignment on Six Scientific Domains"
_ICLR.cc/2026/Conference — ICLR 2026 Poster_

### Official Review · Reviewer_GZL6 · 2025-10-25

**Soundness:** 2
**Presentation:** 3
**Contribution:** 2
**Rating:** 2
**Confidence:** 3

**Summary:**

LLMs are advancing in complex tasks but their resilience against misuse, especially scientifically sophisticated risks, is underexplored. Existing safety benchmarks focus on low-risk instructions or minimal knowledge comprehension, failing to assess model safety in knowledge-intensive scenarios. SOSBENCH, a regulation-grounded benchmark, aims to address this gap by incorporating six high-risk scientific domains. However, advanced models consistently disclose disallowed content, highlighting safety alignment deficiencies.

**Strengths:**

+ The authors introduce the first regulation-grounded, hazard-focused benchmark spanning six scientific disciplines.

**Weaknesses:**

- The scope of the benchmark is too large (scientific knowledge), and the many domain knowledge is not involved.
- Only a few jailbreaking methods are involved, which hinder the completeness of the safety analysis.
- The benchmark only contain a few models in different domains, and even some of the representative models are missing.

**Questions:**

1. The benchmark data is generated intrinsically by LLMs, which means the data could be too synthetic. The reliability should be examined by the experts.

2. The scope of the benchmark is too large (scientific knowledge), and many domain knowledge areas are not involved. For example, toxicity in chemistry and illegal drugs in medicine. A lot of knowledge in different domains is missing. I would recommend that the authors narrow the scope.

3. The experimental analysis only involves a limited number of LLMs. Some representative domain-specific models, e.g., BioGPT and MedPaLM, are not included. This hinders the reliability of the conclusion.

---

> ### Author Response · Authors · 2025-11-18
> **Response**
>
> ## Our scope
> > The scope of the benchmark is too large (scientific knowledge), and the many domain knowledge is not involved.
> > The scope of the benchmark is too large (scientific knowledge), and many domain knowledge areas are not involved. For example, toxicity in chemistry and illegal drugs in medicine. A lot of knowledge in different domains is missing. I would recommend that the authors narrow the scope.
>
>
>
> We respectfully disagree with this assessment.
>
>
> **The mentioned missing examples ARE covered with grounding:**
>
> 1. **"chemistry"**: Our chemistry domain uses NFPA 704 Level 4 hazards—the highest classification. See Section 3.1, Table 12.
> 2. **"medicine"**: We have an **entire pharmacology domain** (500 samples) covering controlled substances, grounded in DEA regulations (21 CFR §1308.13) and NIDA drug lists. See Table 2: pharmacology shows the **highest PVR** (85.0% for GPT-4.1), demonstrating this is a critical vulnerability.
>
> **Why 6 domains is appropriate scope:**
>
> - **"Scientific knowledge" is the established scope for this research area** (See Table 1):
>   - We call it a scientific knowledge benchmark because it benchmarks the safety related to scientific knowledge, but not else, our benchmark evaluates safety alignment on **scientific knowledge** across multiple high-risk domains
>   - Compare to prior work in this area:
>     - SciMT-Safety ("Control risk for potential misuse of artificial intelligence in **science**"): 2 domains
>     - SciSafeEval ("Safety alignment of large language models in **scientific** tasks"): 4 domains
>     - **SOSBench (Ours)**: 6 regulation-grounded domains
>   - No benchmark can cover "all" scientific knowledge (impossible) as there are always new terms even new scientific domains, but ours provides the **most comprehensive coverage to date** within this established research scope
> - **Each domain is regulation-grounded**, not arbitrary (Table 12: NFPA, DEA, WHO ICD-11, IAEA, DSM-5)
> - **Our Finding 2**: "Alignment on pharmacology is particularly shallow"—we'd miss this with narrow scope
> - **Real-world deployment**: LLMs are general-purpose; comprehensive evaluation is essential
>
> Narrowing scope would eliminate coverage of critical areas and weaken our contribution.
>
>
> ## Jailbreak Test
> > Only a few jailbreaking methods are involved, which hinder the completeness of the safety analysis.
>
> **Jailbreak is not the primary focus of our benchmark.**
>
> As noted at the beginning of Finding 8, the jailbreak experiments are not the primary focus of our safety benchmark, but an exploratory study to further probe the robustness of safety alignment and to illustrate a potential downstream application of SoSBench. We therefore intentionally use a small set of representative jailbreak methods, which we find adequate for this purpose; adding many more would increase space and complexity without substantially changing the conclusions of our safety analysis.
>
>
> ## Model Set
> > The benchmark only contain a few models in different domains, and even some of the representative models are missing.
> > The experimental analysis only involves a limited number of LLMs. Some representative domain-specific models, e.g., BioGPT and MedPaLM, are not included. This hinders the reliability of the conclusion.
>
> **26 general-purpose LLMs are included in main evaluation**
>
> We respectfully disagree that 26 models constitutes a "limited number." This represents one of the most comprehensive model evaluations in the safety benchmarking literature. Our evaluation spans, 7 major model families(GPT, Gemini, Claude, Llama, Qwen, Deepseek, Grok), with open and closed-source models across multiple scales and reasoning/non-reasoning paradigms.
>
>
> **7 domain models are evaluated**
>
> We already cover a diverse set of strong, domain-specialized models, including`BioMistral-7B-SLERP`, `ChemDFM-v1.5-8B`, `Med-LLaMA3-8B`, `PsychoCounsel-Llama3-8B`, `Llama3.1-70B-ShiningValiant2`, and `Intern-S1`. These models spin different domains, model sizes, and developers. We therefore believe the trends we report are representative of current practice.
>
> **Suggested models are inappropiate**
>
> BioGPT (GPT-2–based, 0.4B, developed 2022) is considerably too small and has significantly been superseded by the newer architectures and larger size we include, so its addition would add limited insight beyond serving as an outdated baseline. MedPaLM (released in 2023), on the other hand, does not provide public access, so we are unable to evaluate it.

---

> > ### Author Response · Authors · 2025-11-18
> > **Response (Continued)**
> >
> > ## Data Quality
> > > The benchmark data is generated intrinsically by LLMs, which means the data could be too synthetic. The reliability should be examined by the experts.
> >
> >
> > **Our pipeline is LLM-assisted, not purely synthetic, and is filtered by expert validation.**
> > We clarify that SoSBench is not purely LLM-generated: we start from hand-crafted, regulation-grounded seed scenarios and only then use LLMs to expand/paraphrase them. And LLM-assisted data synthesis with subsequent human filtering is standard practice in safety benchmarking [1–2], and in our case domain experts manually inspect and validate the data to ensure factual correctness, realism, and clarity.
> >
> > [1] Souly A, Lu Q, Bowen D, Trinh T, Hsieh E, Pandey S, Abbeel P, Svegliato J, Emmons S, Watkins O, Toyer S. A strongreject for empty jailbreaks. Advances in Neural Information Processing Systems. 2024 Dec 16;37:125416-40.
> >
> > [2] Jiang L, Rao K, Han S, Ettinger A, Brahman F, Kumar S, Mireshghallah N, Lu X, Sap M, Choi Y, Dziri N. Wildteaming at scale: From in-the-wild jailbreaks to (adversarially) safer language models. Advances in Neural Information Processing Systems. 2024 Dec 16;37:47094-165.
> >
> >
> > # Thank you
> > We sincerely appreciate your thoughtful feedback. If any points remain unclear, we welcome further questions and look forward to continuing the discussion. We also hope our explanations merit reconsideration and a potential increase in the evaluation score of our work.

---

> > > ### Author Response · Authors · 2025-11-27
> > > **Follow-up**
> > >
> > > Dear Reviewer GZL6,
> > >
> > > Thank you again for the time and effort you’ve already invested in assessing our submission.
> > >
> > > As the ICLR discussion period is approaching its end, we would greatly appreciate any further feedback or questions. We have provided detailed responses and implemented several revisions based on the initial comments. Two reviewers (KEhQ, jujf) have indicated that they are satisfied with our responses and have updated their scores. We hope our revisions also address your concerns and that you might consider reassessing our work.
> > >
> > > We fully understand that everyone is busy, but if there are remaining concerns or points that are unclear, we would be grateful for the opportunity to address them before the discussion phase closes.
> > >
> > > Best regards,
> > > Authors

---

> > > > ### Comment · Reviewer_GZL6 · 2025-11-28
> > > >
> > > > Sorry for the late reply! Thank you for your rebuttal. I would like to increase my score. However, I still think the scope is too large.

---

> ### Author Response · Authors · 2025-11-28
>
> Thank you again for taking the time to reassess our work.
>
> ### Response to last concern of the scope
>
> In response to this, we have revised our title to *SOSBENCH: Benchmarking Safety Alignment on **Six Scientific Domains*** and updated the abstract and introduction to explicitly state that our contribution focuses on these six scientific domains. We also now highlight the extension to additional domains as future work (L490–492 in the revised manuscript).
>
> We view this primarily as a clarification of framing rather than a change to the core methodology or findings: the benchmark, analyses, and main conclusions remain unchanged and already operate within this six-domain scope. We hope these revisions help address the last concern about scope, and we are very grateful once again for your careful assessment and feedback.

---

### Official Review · Reviewer_qh3U · 2025-10-26

**Soundness:** 3
**Presentation:** 3
**Contribution:** 3
**Rating:** 4
**Confidence:** 4

**Summary:**

This paper introduces SoSBench, a regulation-grounded benchmark designed to evaluate the safety alignment of large language models (LLMs) across six high-risk scientific domains, including chemistry, biology, medicine, pharmacology, physics, and psychology. The benchmark is constructed through a three-stage process: expert seed term extraction from regulations, template-based prompt generation, and LLM-assisted evolution using coverage-driven sampling. Evaluation across multiple LLMs under varying reasoning budgets and jailbreak settings reveals that advanced models still exhibit unsafe behaviors in domain-specific high-risk contexts.

**Strengths:**

1. The task is significant: existing safety benchmarks mainly focus on generic or shallow risks, leaving gaps in evaluating scientifically grounded hazards defined by formal regulations.

2. The data construction pipeline is novel and structured, and the inclusion of a Lite version lowers the barrier to reproduction and large-scale testing.

3. The experiments are comprehensive, covering diverse LLMs and testing configurations, and yield meaningful insights into safety behavior patterns.

**Weaknesses:**

1. Quality verification only checks whether responses are harmful, not whether the questions themselves are rigorous or unambiguous. There is no reported validation of problem correctness or semantic clarity.

2. Hill notation (mentioned in Section 3.2.1) and similar representations may not be unique identifiers; multiple structural isomers can share the same formula, leading to ambiguous or misleading prompts.

3. The paper does not convincingly demonstrate that SoSBench-Lite results are consistent with the full benchmark; no statistical correlation or ranking consistency is reported.

4. The paper provides limited constructive guidance on improving safety alignment—no concrete methods are proposed to mitigate the identified vulnerabilities.

5. Although the appendix discusses the use of an LLM judge, relying on a single model as evaluator could introduce bias; stronger or more diverse judges might produce different results.

The paper tackles a highly relevant and underexplored problem and contributes a potentially valuable benchmark. However, key methodological and validation issues remain unresolved—particularly question quality assurance, data transparency, and the lack of quantitative verification for SoSBench-Lite’s representativeness. Addressing these could elevate the work to acceptance level.

**Questions:**

1. How can the authors verify that the generated questions are of high quality—for example, logically sound and factually valid?

2. In Section 3.2.1, the paper mentions using alternative forms to replace core terms; how do the authors ensure one-to-one correspondence (e.g., molecular Hill formulas not mapping to multiple isomers)?

3. What empirical evidence supports that the Lite benchmark can reliably approximate results from the full dataset?

4. Could the authors disclose more details about the Manual Seed Collection phase—specifically, the number and background of experts, term-pool size, and review/adjudication workflow?

5. Regarding Finding 5, why would a model with visible CoT become more unsafe simply by increasing reasoning budget? If the CoT were hidden and only final answers shown, would the same issue persist?

---

> ### Author Response · Authors · 2025-11-18
> **Response**
>
> ## Quality Verification & Control
> > Quality verification only checks whether responses are harmful, not whether the questions themselves are rigorous or unambiguous. There is no reported validation of problem correctness or semantic clarity.
>
> > How can the authors verify that the generated questions are of high quality—for example, logically sound and factually valid?
>
> **Question quality is already rigorously vetted before the stage that evaluates model harmfulness.**
> We agree that the rigor and clarity of the questions themselves are important, and we apologize that our current version underemphasizes this aspect. During dataset construction, all scenarios were authored from concrete, real-world regulations and guidelines and then manually reviewed by domain annotators for factual correctness and semantic clarity. Ambiguous or ill-posed items were edited or discarded through multiple iterations. The quality verification stage we describe is therefore in addition to this earlier vetting and is focused specifically on the harmfulness and policy violation of model outputs.
>
> ## Hill notation clarification
> > Hill notation (mentioned in Section 3.2.1) and similar representations may not be unique identifiers; multiple structural isomers can share the same formula, leading to ambiguous or misleading prompts.
>
> > In Section 3.2.1, the paper mentions using alternative forms to replace core terms; how do the authors ensure one-to-one correspondence (e.g., molecular Hill formulas not mapping to multiple isomers)?
>
> **Ambiguous forms like Hill notation are used to test whether models still behave safely.**
>
> A key goal of Section 3.2.1 is to test whether a model treats semantically equivalent prompts consistently. Real users often reference chemicals with incomplete or ambiguous terms, and Hill notation is one such ambiguity observed in practical misuse scenarios. Including this form strengthens robustness testing rather than weakening it. From a safety perspective, an LLM should remain cautious even when a term is close to, but not identical with, a harmful entity. In our benchmark, all unambiguous identifiers already map to unique safety labels, and the addition of partially ambiguous surface forms allows us to evaluate whether models maintain appropriate caution in borderline cases.
>
>
> ## Manual Seed Collection Details
> > Could the authors disclose more details about the Manual Seed Collection phase—specifically, the number and background of experts, term-pool size, and review/adjudication workflow?
>
>
> **Seed terms and prompts were drafted and vetted through a multi-stage process led by domain experts.**
>
> The expert team consists of faculty-level and graduate-level researchers with specialized training in the relevant scientific areas. Each annotator independently extracted and mined seed terms from the relevant regulations, laws, and standards in each domain. Chemical, pharmacy, and physics terms were further expanded with alternative names and variations obtained from external knowledge bases. In total, we collected **7682** seed terms across six domains. Annotators manually wrote the seed prompts, and LLM-generated prompt variations were reviewed to ensure correctness and risk fidelity. After the data evolution stage, we conducted a second review pass to verify term usage and prompt quality. To enhance evaluator performance, we manually annotated the seed set as safe, unsafe, or unsure following the labeling rules described in Appendix A.2. Prompts containing clearly harmful intent, including detailed instructions, experimental procedures, advice against professional medical care, deliberate spread of infectious agents, or unsafe practices, were labeled unsafe, while refusals, harm mitigating advice, and safe practice guidance were labeled safe. Further procedural details are provided in Appendix A.2 and Appendix C.

---

> > ### Author Response · Authors · 2025-11-18
> >
> > ## SOSBench-Lite Representativeness
> > > The paper does not convincingly demonstrate that SoSBench-Lite results are consistent with the full benchmark; no statistical correlation or ranking consistency is reported.
> >
> > > What empirical evidence supports that the Lite benchmark can reliably approximate results from the full dataset?
> >
> >
> > **SoSBench-Lite closely matches the full benchmark.**
> >
> > We have added a consistency analysis between the full SoSBench and SoSBench-Lite (Section 3). Across all evaluated models, the Lite suite shows very high agreement with the full benchmark, with Pearson correlation coefficient $r=0.997$  and Spearman $\rho=0.990$. These results empirically support that SoSBench-Lite can reliably approximate the rankings and overall trends obtained from the full dataset.
> >
> >
> >
> >
> > ## Guidance on improving safety alignment
> > > The paper provides limited constructive guidance on improving safety alignment—no concrete methods are proposed to mitigate the identified vulnerabilities.
> >
> > **Our main contribution is the benchmark, complemented by a pilot unlearning-based mitigation study.**
> > We agree on the importance of constructive guidance for safety alignment. To this end, we include a pilot unlearning-based mitigation study (Finding 6), which explicitly demonstrates a safety–utility trade-off and illustrates how SoSBench can be used to stress-test candidate defenses. That said, instead of focusing on proposing new concrete alignment approach, our primary contribution is the construction and evaluation of SoSBench itself, which we intend as a foundation and proxy for future work to design and rigorously compare concrete alignment methods.
> >
> >
> >
> > ## LLM as Judge Design
> > > Although the appendix discusses the use of an LLM judge, relying on a single model as evaluator could introduce bias; stronger or more diverse judges might produce different results.
> >
> > **We mitigate single-judge bias via human calibration and a strong judge model.**
> > Using single judge LLM is a common practice in this research field. And we try our best to mitigate the risk of single-judge bias by calibrating the LLM judge against human annotations and only adopting it after confirming high agreement. Based on this calibration study, we select `GPT-5` as our judge model, which is the strongest model available at the time of our development. While we acknowledge that multi-judge or ensemble setups might further reduce bias, they introduce substantial design complexity and cost (especially for reasoning models), so we leave them as valuable future extensions beyond our current work.
> >
> >
> > ## Finding 5 Interpretation
> > > Regarding Finding 5, why would a model with visible CoT become more unsafe simply by increasing reasoning budget? If the CoT were hidden and only final answers shown, would the same issue persist?
> >
> >
> > With CoT visible, the model expose full internal thinking to user (even potetntial misuse adversary), and regardless whether final answer is safe or not, the longer CoT content provide more space to carry potential policy-violation content, even the model is just 'thinking' or 'planning' answer. For example, when model finally refuse to answer instruction of explosive components, it is possible that the model think of the process  firstly, which is policy violation indeed.
> >
> >
> > # Thank you
> > We sincerely appreciate your thoughtful feedback. If any points remain unclear, we welcome further questions and look forward to continuing the discussion. We also hope our explanations merit reconsideration and a potential increase in the evaluation score of our work.

---

> > > ### Author Response · Authors · 2025-11-27
> > > **Follow-up**
> > >
> > > Dear Reviewer qh3U,
> > >
> > > Thank you again for the time and effort you’ve already invested in assessing our submission.
> > >
> > > As the ICLR discussion period is approaching its end, we would greatly appreciate any further feedback or questions. We have provided detailed responses and implemented several revisions based on the initial comments. Two reviewers (KEhQ, jujf) have indicated that they are satisfied with our responses and have updated their scores. We hope our revisions also address your concerns and that you might consider reassessing our work.
> > >
> > > We fully understand that everyone is busy, but if there are remaining concerns or points that are unclear, we would be grateful for the opportunity to address them before the discussion phase closes.
> > >
> > > Best regards,
> > > Authors

---

### Official Review · Reviewer_q51o · 2025-10-31

**Soundness:** 2
**Presentation:** 3
**Contribution:** 2
**Rating:** 4
**Confidence:** 4

**Summary:**

The manuscript presents SOSBENCH, a 3,000-prompt, regulation-grounded benchmark for measuring LLM safety in six domains (chemistry, biology, medicine, pharmacology, physics, psychology). Prompts are seeded from regulatory terminology, expanded via an LLM-assisted evolutionary pipeline, and validated with surrogate models. The authors evaluate a wide range of frontier models and report Policy Violation Rate (PVR) per model/domain using an LLM-as-judge pipeline. Results indicate high rates of policy-violating output across many models and domains.

**Strengths:**

1. The paper addresses a timely and important problem by evaluating model safety in scientific domains where domain-specific knowledge can lead to misuse. This fills a clear gap in existing safety benchmarks and has high potential impact for both research and deployment.
2. The work is well grounded in authoritative regulatory sources. This regulatory foundation enhances real-world relevance and helps ensure that the benchmark reflects authentic, high-risk scenarios rather than artificial examples.
3. The experimental evaluation is extensive and thorough. The authors test a wide range of both open and closed models, report per-domain Policy Violation Rates (PVRs), and include adversarial multi-turn experiments that effectively demonstrate model brittleness.

**Weaknesses:**

1. Given that all safety scores rely entirely on GPT-5 as the evaluator, how reliable is this judgment process in practice? Have the authors provided quantitative validation against human annotations, for example, agreement metrics such as Cohen’s κ or F1 on a human-labeled subset, and discussed potential biases or failure cases of the judge?
2. The multi-turn jailbreak experiments are intriguing, but can the authors clarify the exact setup? What specific attack algorithms, interaction lengths, and numbers of trials were used, and how can these results be reproduced independently?
3. Several baselines in Table 1 seem to be taken from prior papers that used different evaluation settings. Have any baselines been re-evaluated under the same GPT-5 judging and decoding setup, or are the incomparable results clearly marked as such?
4. Line 229 states “LlamaGuard 2” while footnote links to Llama-Guard-3-8B, please clarify exact versions.

**Questions:**

1. Beyond binary PVR, do you report a severity-weighted metric (mapped to specific clauses/levels in the cited standards) so minor slips don’t count the same as high-risk procedural guidance?
2. Did the author check for overlap between SOSBENCH prompts and public training/eval corpora, and quantify any leakage that could inflate or skew results?

---

> ### Author Response · Authors · 2025-11-18
> **Response**
>
> ## LLM Judge Reliability
> > Given that all safety scores rely entirely on GPT-5 as the evaluator, how reliable is this judgment process in practice? Have the authors provided quantitative validation against human annotations, for example, agreement metrics such as Cohen’s κ or F1 on a human-labeled subset, and discussed potential biases or failure cases of the judge?
>
> **GPT-5’s safety judgments are quantitatively calibrated against human annotations.**
>
>
> We do provide a quantitative human-annotation study to calibrate GPT-5 as the safety judge. As described in Sec. 4.1 (original L371–373) and detailed in Appendix A.2, we compare multiple candidate judge models against human-labeled subsets and report standard agreement metrics (e.g., F1 / PCC), and find that our final GPT-5 setting achieves the strongest consistency with human annotators.
>
>
>
>
>
> ## Multi-turn Jailbreak
> > The multi-turn jailbreak experiments are intriguing, but can the authors clarify the exact setup? What specific attack algorithms, interaction lengths, and numbers of trials were used, and how can these results be reproduced independently?
>
> **Updated multi-turn jailbreak setup for reproducibility.**
>
> We clarify that our multi-turn jailbreak experiments use the Crescendo attack from the Microsoft PyRIT library, with `GPT-4.1` as both the adversary and scoring model to attack the victim models. We set `max_turns = 5` and `max_backtracks = 2`. We have added these implementation details to the revised paper so that the experiments can be independently reproduced.
>
>
>
>
> ## Table 1 Result
> > Several baselines in Table 1 seem to be taken from prior papers that used different evaluation settings. Have any baselines been re-evaluated under the same GPT-5 judging and decoding setup, or are the incomparable results clearly marked as such?
>
> We use `GPT-5` judge to evaluate model response used in Table 1.
>
> ## LlamaGuard Notation
> > Line 229 states “LlamaGuard 2” while footnote links to Llama-Guard-3-8B, please clarify exact versions.
>
> We clarify that we use **Llama Guard 3**, as correctly indicated by the footnote link to `Llama-Guard-3-8B`. In Line 229, the “2” after *LlamaGuard* is a superscript citation marker rather than a version number.
> We revised the text to explicitly write “LlamaGuard3” to avoid any ambiguity.
>
>
>
> ## Evaluation Metric
> > Beyond binary PVR, do you report a severity-weighted metric (mapped to specific clauses/levels in the cited standards) so minor slips don’t count the same as high-risk procedural guidance?
>
> **We prioritize a unified cross-domain metric**
> We appreciate this helpful suggestion. In the current work, we deliberately focus on a unified binary PVR metric that is comparable across domains and standards. While a severity-weighted metric aligned with specific clauses/levels would indeed provide more fine-grained insight, it is challenging to design a single, principled weighting scheme that remains consistent across heterogeneous domains and regulatory frameworks. Adding clause-level, severity-aware scoring will be an important direction for future extensions of SOSBench.
>
>
>
>
> ## Benchmark Overlap
> > Did the author check for overlap between SOSBENCH prompts and public training/eval corpora, and quantify any leakage that could inflate or skew results?
>
> **No overlap founded**
> For the models we evaluate, there is no transparent disclosure of their full training corpora, so we cannot reliably audit data overlap in a quantitative way. Regarding evaluation corpora, to the best of our knowledge SOSBench is the first benchmark to target this specific regulation-grounded safety setting, and our literature review did not reveal any existing evaluation datasets with overlapping prompts.
>
>
>
> # Thank you
> We sincerely appreciate your thoughtful feedback. If any points remain unclear, we welcome further questions and look forward to continuing the discussion. We also hope our explanations merit reconsideration and a potential increase in the evaluation score of our work.

---

> > ### Comment · Reviewer_q51o · 2025-11-18
> > **Thank you!**
> >
> > Many thanks to the authors for the detailed responses and clarifications, which resolves the previous questions. For "Adding clause-level, severity-aware scoring will be an important direction for future extensions of SOSBench." in your response, could we kindly update this in the future work part of the Conclusion section?
> >
> > Thanks again and I have increased soundness score by 1 and overall rating by 2.

---

> > > ### Author Response · Authors · 2025-11-18
> > >
> > > Thank you very much for your engagement and for raising the assessment! We are glad that our responses resolved your previous concerns.
> > >
> > >  Per your advice, we updated the "Future Work" paragraph in the Conclusion section to explicitly highlight the development of clause-level, severity-aware scoring as a critical direction for future work.
> > >
> > > Thank you again for your constructive feedback which has helped improve the quality of our work.

---

### Official Review · Reviewer_KEhQ · 2025-10-31

**Soundness:** 3
**Presentation:** 4
**Contribution:** 2
**Rating:** 4
**Confidence:** 4

**Summary:**

The paper highlights safety alignment deficiencies in a wide range of advanced LLMs. This is demonstrated on a collection of scientific prompts that elicit unsafe behaviour from the models. The set of prompts is collected by conditioning on regulatory text and it is constructed in a manner to increase coverage of scenarios.

**Strengths:**

- The paper highlights an important issue raised in other forms by various other works, that general safety mechanisms have blind spots on domain-specific knowledge.
- The provided benchmark and empirical analysis highlight this shortcoming well. The empirical analysis in particular is very comprehensive, evaluating a wide range of model classes and scales.

**Weaknesses:**

- The claimed difference in hazard level from SciSafeEval [1] is not as pronounced as suggested. Examining the example prompts from either benchmark in Appendix B.2, a person intending to act on the model’s output is equally likely to pose either question. What, then, is the precise aspect of hazard level that this benchmark seeks to distinguish? Clarifying this distinction would help future practitioners determine which benchmark is most appropriate for their specific use cases. A related concern is raised below.
- The overall motivation for evaluating on this benchmark would also benefit from additional clarity. It appears to target safety vulnerabilities in scenarios where the model’s scientific knowledge is being applied: yet, as stated in L49–50, not in cases involving “advanced” scientific knowledge. This distinction is somewhat ambiguous and should be elaborated.
- *Missing standard error in the empirical analysis:* The results in Tables 2 and 3 and Figure 3 are averaged over multiple data points and categories. The underlying variance or standard error of the PVR should be reported to assess statistical significance. For instance, the difference in PVR for Finding 5 (as supported by Figure 5) is quite small—on the order of 0.1 across budgets. Reporting the standard error in such cases is necessary to gauge the statistical strength of the results.

---
[1] Tianhao Li, Jingyu Lu, Chuangxin Chu, Tianyu Zeng, Yujia Zheng, Mei Li, Haotian Huang, Bin
Wu, Zuoxian Liu, Kai Ma, et al. Scisafeeval: a comprehensive benchmark for safety alignment
of large language models in scientific tasks. arXiv preprint arXiv:2410.03769, 2024b.

**Questions:**

What does Figure 3 look like if SciSafeEval is also included? It would be insightful to show a comparison to the most similar existing benchmark.

---

> ### Author Response · Authors · 2025-11-18
> **Response**
>
> ## Compare to SciSafeEval
> > The claimed difference in hazard level from SciSafeEval [1] is not as pronounced as suggested. Examining the example prompts from either benchmark in Appendix B.2, a person intending to act on the model’s output is equally likely to pose either question. What, then, is the precise aspect of hazard level that this benchmark seeks to distinguish? Clarifying this distinction would help future practitioners determine which benchmark is most appropriate for their specific use cases. A related concern is raised below.
>
> > What does Figure 3 look like if SciSafeEval is also included? It would be insightful to show a comparison to the most similar existing benchmark.
>
>
> **We distinguish SoSBench by how close compliant answers are to directly executable, regulation-grounded misuse; and compare it to SciSafeEval in Figure 3.**
>
>
> We thank the reviewer for this helpful question. Instead of meaning SciSafeBench is not hazardous at all, our notion of “hazard level” focuses how close a fully compliant answer is to **operational, ready-to-execute misuse**. As illustrated in Table 11 and Appendix B.2, SciSafeEval prompts request knowledge retrieval or classification (e.g., identifying reactants), so compliant answers remain non-actionable hints that still require substantial expertise and additional steps to cause harm. By contrast, SOSBench prompts are explicitly grounded in concrete regulations and solicit multi-step device/protocol-level guidance whose literal execution would directly constitute harmful practice.
>
> In the example of Table 11, a non-refusal answer to the SciSafeEval prompt provides information yet remains steps away from immediate practical harm, whereas an answer to the SOSBench prompt is closer to an actionable, harmful procedure.
>
> Also we have updated Figure 3 in the revision to also include SciSafeEval (300 randomly sampled instructions)for comparison with SoSBench.
>
>
> ## Motivation Clarification
> > The overall motivation for evaluating on this benchmark would also benefit from additional clarity. It appears to target safety vulnerabilities in scenarios where the model’s scientific knowledge is being applied: yet, as stated in L49–50, not in cases involving “advanced” scientific knowledge. This distinction is somewhat ambiguous and should be elaborated.
>
>
> **SoSBench targets non-trivial applied scientific knowledge in high-risk settings.**
>
> We apologize for the confusion. To clarify, our intent was not to downplay “advanced” science, but to emphasize that SoSBench focuses on *applied scientific knowledge* in regulation-grounded, high-risk scenarios, typically requiring university-level or graduate-level expertise to understand and answer.
>
>
> This is reflected in our difficulty analysis (Table 4–5): 72% of prompts are rated medium–hard in comprehension difficulty and 91% are medium–hard in answering difficulty, indicating that most scenarios require non-trivial scientific background rather than simple factual recall.
>
>
>
> ## Statistical Analysis
> > Missing standard error in the empirical analysis: The results in Tables 2 and 3 and Figure 3 are averaged over multiple data points and categories. The underlying variance or standard error of the PVR should be reported to assess statistical significance. For instance, the difference in PVR for Finding 5 (as supported by Figure 5) is quite small—on the order of 0.1 across budgets. Reporting the standard error in such cases is necessary to gauge the statistical strength of the results.
>
> **We updated confidence intervals with PVR.**
>
> To better support statistical analysis but maintain readability, we have updated the overall evaluations in Tables 2 and 3 to include confidence intervals. For Figure 3 in Finding 5, we also add the corresponding intervals (error bars). Please see the updated version.
>
>
> # Thank you
> We sincerely appreciate your thoughtful feedback. If any points remain unclear, we welcome further questions and look forward to continuing the discussion. We also hope our explanations merit reconsideration and a potential increase in the evaluation score of our work.

---

> > ### Comment · Reviewer_KEhQ · 2025-11-20
> >
> > Thanks for the clarifications. It appears that Figures 3 and 5 have not been updated as described in the response -- this may have been overlooked. Could you please update the pdf with the revised figures so that the responses are better contextualized.
> >
> > > confidence intervals in Tables 2 and 3
> >
> > This seems underspecified. Could you clarify which type of confidence intervals were used? I am unable to find this information in the current pdf.

---

> > > ### Author Response · Authors · 2025-11-20
> > > **Follow-up Response**
> > >
> > > ### Updated Figures
> > > We apologize for using incorrect file names in our last revision, which omitted the updated Figures 3 and 5 as described. We have now updated our paper with the corrected figures, so the changes (including the added intervals and SciSafeEval comparison) are properly reflected and contextualized in the main text.
> > >
> > >
> > > ### Added Specification on Confidence Interval
> > >
> > > For Tables 2 and 3, we treat the underlying judge results as samples, compute sample variance $s^2$, and report the confidence interval $CI = z_{1-\alpha/2}\, s\, m^{-1/2}$, where $\alpha = 0.1$ and $s$ is the standard deviation.
> > >
> > > We added this specification to the paper for clarity.
> > >
> > > ## Thank you
> > > We sincerely appreciate your feedback, which helps improve the quality of our paper. We hope our explanations merit reconsideration and a potential increase in the evaluation score of our work. We welcome further questions and look forward to continuing the discussion if any points remain unclear.

---

> > > > ### Comment · Reviewer_KEhQ · 2025-11-21
> > > >
> > > > Thanks for the response, I will update my score. Regarding the updates, a few questions arise:
> > > > - As per the updated Figure 3: SciSafeEval covers topics that are different than those in SOSBench. How can that be the case, given Table 1 shows than it should cover 4 our of 6 of the same topics? And why is the coverage so narrow, even smaller than other benchmarks in Table 1 that are less diverse?
> > > > - As per the updated Figure 5: does Finding 5 still hold? The PVRs for reasoning effort scaling lie within each other's confidence intervals. So can the claim still hold?

---

> > > > > ### Author Response · Authors · 2025-11-22
> > > > >
> > > > > ### Q1: SciSafeEval Diversity
> > > > > We appreciate the reviewer’s insightful interpretation of Figure 3, which visualizes the semantic distribution of individual prompts/instructions. SciSafeEval indeed covers 4 of the 6 domains that SoSBench targets (that's why SciSafeEval overlaps tightly with SOSBench on some regions), but its scientific items focus on designed subtasks (e.g., molecule generation, property prediction, reaction prediction in chemistry) using homogeneous instructions with various terms, leading to potential **semantic clusters** in the embedding space. SoSBench is explicitly designed to broaden semantic diversity within each domain, which produces a wider spread.
> > > > >
> > > > > Regarding why SciSafeEval appears narrower than some “less diverse” benchmarks in Table 1, this reflects differences in **intra-domain variety** rather than domain count alone: those benchmarks with fewer scientific domains nonetheless explore a broader range of instruction styles and misuse scenarios.
> > > > >
> > > > > ### Q2: Finding 5
> > > > > We appreciate the reviewer’s careful check. In Figure 5, we observe that for models with *visible* CoT (Claude-3.7 and Grok-3-mini), the PVR differences across reasoning budgets are larger than the corresponding confidence intervals, so the trend remains statistically meaningful. For models with *invisible* CoT (o4-mini and Gemini-2.5), the PVR changes under budget scaling are small and their confidence intervals slightly overlap, which is why we described the effect as “potentially lowering PVR” in our original manuscript.
> > > > >
> > > > > To better reflect this, we now treat the effect for invisible-CoT models as suggestive rather than strong, and we have revised Finding 5 to: **“Test-time scaling harms the safety of visible-thinking models, but slightly benefits that of invisible-thinking models.”** We have also rephrased the analysis in Finding 5 accordingly.
> > > > >
> > > > >
> > > > >
> > > > > ### Thank you
> > > > > The discussion has been greatly helpful in strengthening our manuscript, and we sincerely appreciate your constructive feedback. We look forward to continuing the discussion if any points remain unclear.

---

> > > > > > ### Comment · Reviewer_KEhQ · 2025-11-24
> > > > > >
> > > > > > Thank you for the response. I have updated by score.

---

> > > > > > > ### Author Response · Authors · 2025-11-25
> > > > > > >
> > > > > > > Thank you very much for your thoughtful assessment and for taking the time to engage with our clarifications and revisions. We appreciate your careful reading and feedback, which have helped us improve our work.

---

### Author Response · Authors · 2025-11-18
**Global Response**

We sincerely appreciate the feedback from all reviewers. And we are delight to see our work is acknowledged by all reviewers for our contribution in AI safety research as follows:

**1. Addresses a Critical and Timely Problem**
- **Reviewer KEhQ**: Our work highlights "an important issue... that general safety mechanisms have blind spots on domain-specific knowledge"
- **Reviewer q51o**: This work "fills a clear gap" with "high potential impact for both research and deployment"
- **Reviewer qh3U**: "The task is significant" as existing benchmarks focus on "generic or shallow risks"

**2. Novel, Regulation-Grounded Methodology**
- **Reviewer GZL6**: We introduce the "**first** regulation-grounded, hazard-focused benchmark spanning six scientific disciplines"
- **Reviewer q51o**: Our benchmark is "well grounded in authoritative regulatory sources" with "real-world relevance"
- **Reviewer qh3U**: "The data construction pipeline is novel and structured"

**3. Comprehensive and Rigorous Evaluation**
- **Reviewer KEhQ**: "The empirical analysis... is **very comprehensive**"
- **Reviewer q51o**: Our evaluation is "extensive and thorough... covering both open and closed models"
- **Reviewer qh3U**: The experiments "yield meaningful insights into safety behavior patterns"

We have provided detailed responses to each reviewer's questions and concerns in separate replies below.

---

### Meta-Review · Area_Chair_WwHN · 2026-01-05

**Summary:**

The paper proposes the SOSBench, a regulation-grounded, hazard-focused benchmark including many knowledge-intensive domains. Compared with former benchmarks, this paper focuses on more complex and harmful problems. And comprehensive evaluations show that many LLMs perform unsatisfying on these domains like GPT-4.1, which can alarm the whole community.

The paper proposed a new but important view on safety evaluations, that is, harmful questions with complex knowledge. And the weaknesses they found for current large LLMs can have a large impact on the safety domain to allowing researchers to tackle the safety gaps in complex domains. Most reviewers show a positive attitude before the event. And I think their concerns are well addressed. Therefore, I suggest accepting.

**Reviewer Concerns:**

The concerns mainly lie in the data quality, the evaluation method, and writing. I think they are well-addressed in the rebuttal.

**Reviewer Scores:**

I think the reviewers' score will be increased as the concerns are well-addressed. Before the event, 3 reviewers demonstrate their willingness to increase their score.

---

### Decision · Program_Chairs · 2026-01-26

Accept (Poster)